

# A stable ultrastructural pattern despite variable cell size in *Lithothamnion corallioides*

Valentina Alice Bracchi[1], Giulia Piazza[1], Daniela Basso[1]

[1]Department of Earth and Environmental Sciences, University of Milano-Bicocca, Milan, 20126, Italy
*Correspondence to*: Valentina A. Bracchi (valentina.bracchi@unimib.it)

**Abstract.** Recent advances on the mechanism and pattern of calcification in coralline algae lead to contradictory conclusions.
Coralline calcification appears biologically induced, as suggested by the dependency of its elemental composition on environmental variables. However, evidence of a biologically controlled calcification process, resulting in distinctive patterns at the scale of family, was also observed. In order to clarify the matter, five collections of *Lithothamnion corallioides* from the Atlantic Ocean and the Mediterranean Sea, across a wide depth range (12-66 m) have been analyzed for morphology, anatomy and cell wall crystal patterns of both perithallial and epithallial cells, in order to detect possible ultrastructural changes. *L.*
*corallioides* shows the alternation of tiers of short-squared and long-ovoid/rectangular cells along the perithallus, forming a typical banding. The perithallial cell length decreases according to water depth and growth-rate, whereas diameter remains constant. Our observations confirm that both epithallial and perithallial cells show primary (PW) and secondary (SW) calcite walls. Rectangular tiles, with the long axis parallel to the cell membrane forming a multi-layered structure, characterize the PW. Flattened squared bricks characterize the SW with roundish outlines enveloping the cell and showing a zigzag pattern.
Long and short cells have different thickness of PW and SW, with a thicker SW and PW in short cells. Epithallial cells are one up to three flared cells, with the same shape of the PW and SW crystals. Despite the diverse seafloor environments and the variable *L. corallioides* growth-rate, the cell walls maintain a consistent ultrastructural pattern, with unaffected crystal shape and arrangement. A comparison with two congeneric species, *L. minervae* and *L. valens*, showed similar ultrastructural patterns in SW, but evident differences in the PW crystal shape. Our observations point to a biological control rather than an induction
of the calcification process in coralline algae, and suggest a possible new morphological diagnostic tool for species identification, with relevant importance for paleontological application. Finally, secondary calcite, in form of dogtooth crystals that fill the cell lumen, has been observed. It represents a form of early diagenesis in living collections which can have implications in the reliability of climate and paleoclimate studies based on the geochemistry techniques.

## 1 Introduction

The subclass Corallinophycidae is spread globally and comprises the crustose coralline algae (CCA), important Mg-calcite calcifiers and habitat builders of rhodolith beds, temperate algal reefs and tropical coralgal reefs (Cabioch and Giraud, 1986;



Adey, 1986; Ries, 2006; Caragnano et al., 2009; Bracchi et al., 2014, 2016). The complex calcifying process in CCA takes place during their whole life span and involves the entire organism. For this reason, corallines bear witness to past benthic primary production by macroalgae with an excellent fossil record (Basso et al., 2007; Bracchi et al., 2014, 2016; Ragazzola et

al., 2020).

Rhodoliths are unattached nodules formed mostly by CCA. Among them, free-living unattached branches usually characterize maerl beds in the NE Atlantic Ocean (Henrich et al., 1995; Birkett et al., 1998; Peña and Bárbara, 2008, 2009; Peña et al., 2014) and in the Mediterranean Sea (Huvé, 1956; Jacquotte, 1962; Gambi et al., 2009; Agnesi et al., 2011; Savini et al., 2012; Basso et al., 2017).

In both geographical settings, the most common species are *Lithothamnion corallioides* (P.Crouan and H.Crouan) P.Crouan and H.Crouan 1867 and *Phymatolithon calcareum* (Pallas) Adey and McKibbin (1970) (Adey and McKibbin, 1970) (Basso et al., 2017; Hernandez-Kantun et al., 2017).

*L. corallioides* is distributed between the Canary Islands, at roughly 28°N, and Scotland, at about 58°N (Irvine and Chamberlain, 1994; Wilson et al., 2004), and is considered one of the most suitable species for paleoclimate reconstruction

(Foster, 2001). This species usually forms twig-like structures, brown to pink or purplish, often sterile, with branch diameters typically in the range 1-2 mm, with knob-like apices (Irvine and Chamberlain, 1994). Primary production, respiration, and calcification of *L. corallioides* are strongly influenced by seasonality, because of the oscillations of temperature and irradiance levels (Adey and Mc Kibbin, 1970; Potin et al., 1990; Martin et al., 2006). *L. corallioides* shows a favorable response to temperature increase, reaching its maximum primary production during summer (Adey and Mc Kibbin, 1970; Potin et al.,

1990). *L. corallioides* minimum survival temperature is between 2°C and 5°C, while the optimal growth is observed between 13°C (Adey and McKibbin, 1970) and 14°C (Blake and Maggs, 2003). In longitudinal sections, the periodical change in growth-rate, due to the alternation of seasons, generates perithallus banding in Lithothamnion species, as in the long protuberances of *L. corallioides*. It results in an evident alternation of tiers of thick-walled, generally short cells versus thin-walled long cells along the main axis of perithallus growth (Basso, 1995; Basso et al., 1997; Kamenos and Law, 2010, Burdett

et al., 2011). The banding has been interpreted as the visible result of the effect of the environment (primarily temperature) on algal growth at different time scales (day, month) (Halfar et al., 2000; Foster, 2001). Alternatively, banding would represent the regular shift between tiers of cells possessing different wall structure (Nash et al., 2019).

In general, the calcifying process of CCA has been described as the deposition of tangential calcite needles in the outer part of the cell wall (interstitial matrix), followed by the formation of radial needles in the cell frame itself in contact with the

plasmalemma. The polysaccharide and fibrillary matrix control both processes (Giraud and Cabioch, 1976; Irvine and Chamberlain, 1994; Adey, 1998; de Cervalho et al., 2017). In *L. corallioides*, calcification has been described as composed of tangential rod-shaped crystals in the primary wall (PW) and perpendicular fan-like rods in the secondary wall (SW) (Giraud and Cabioch, 1976). Borowitzka (1984, 1989) proposed that coralline algae have semi-organized calcification, suggesting that their calcification is biologically controlled, as also indicated by Cabioch and Giraud (1986), rather than induced, as more

recently supported by de Cervalho et al. (2017) and Nash et al. (2019). Cell wall ultrastructures are recognized as a valuable



tool to define the phenotypic expression of genotypic information (Auer and Piller, 2020). The compelling evidence of a biological control over calcification in coralline algae was provided by the identification of family-specific cell wall ultrastructures. In particular, epithallial cells in the genus Lithothamnion show crystal units as thin rectangular blocks (Auer and Piller, 2020). Seasonality, including seawater temperature oscillations and photoperiod, is considered one of the main

factors affecting the growth-rate and the biomineralization process (Steller et al., 2007; Kamenos and Law, 2010; Vásquez-Elizondo and Enríquez, 2017), which may influence the ultrastructure pattern.

The identification of CCA in present-day integrative taxonomy is based on genetic methods coupling with the morphological description and measurement of diagnostic features. Species identification in the fossil record is, on the contrary, merely based on the preservation of morphological taxonomic characters. Consequently, the identification of valuable morphological

characters as a tool for the definition of the paleontological species represents an important challenge. CCA are well represented in the fossil record and *L. corallioides* has been reported in the Pliocene of Spain (Aguirre et al., 2012) and in the Pleistocene of Southern Italy (Bracchi et al., 2014). This study is aimed at describing the ultrastructural mineralogical features of *L. corallioides* from different geographic settings (northeastern Atlantic Ocean and Mediterranean Sea) and across a wide bathymetric interval (12-66 m) to test if the ultrastructural pattern preserves under different environmental conditions and can

be considered as an evident sign of biologically-controlled mineralogical process. Moreover, the identification of specific ultrastructural pattern could be considered as a valuable tool for species identification to be used also in paleobiology. *L. corallioides* has been targeted because of its wide distribution, both geographically and bathymetrically and its occurrence in the fossil record.

## 2 Materials and methods

For this study, we considered five collections (Fig. 1, Tab. 1), from two different geographic settings: the Atlantic Ocean and the Mediterranean Sea, sampled by scuba-diver or grab at different depth ranging from 12 m for Morlaix Bay (France) down to 66 m (Pontine Islands, Italy). Table 1 reports location and date of sampling. All specimens have been collected alive.

In order to highlight possible ultrastructural differences among the same genus, two additional collections, already identified as *Lithothamnion minervae* (Basso, 1995) and *Lithothamnion valens* Foslie 1909 have been considered. These collections have

been sampled alive from Egadi Islands (Italy), during July 2016, at 103 and 86 m of water depth respectively.

### 2.1 Coralline sample preparation

Samples have been prepared for Scanning Electron Microscope (SEM) imaging. Altered, badly preserved or encrusted branches have been discarded. Only the branches showing a shiny surface have been picked from all collections, singly controlled under a Stereo Microscope, and cleaned manually by removing epiphytes and other incrustations. Each sample,

composed of multiple branches, has been cleaned in an ultrasonic bath in distilled water for 10 minutes and air-dried. The branches were then placed in small cylindrical plastic boxes with a base diameter of 1". Branches have been piled up and aligned to obtain multiple layers. The samples were embedded in Epofix resin for SEM analyses, which was stirred for 2 minutes with a hardener (13% w/w), and they were left to harden for one day at room temperature. Samples have then been cut normal to the multiple layers by using a IsoMet diamond wafering blade 15HC, along the direction of branch growth.



Moreover, some additional samples have been prepared for SEM observation, by breaking them with a small chisel. Both longitudinal and surface sections have been selected for SEM observations.

**2.2 Scanning Electron Microscope**

For SEM imaging, the surfaces of embedded samples have been polished by using different sizes of silicon carbide, cleaned ultrasonically in distilled water for 10 minutes and air-dried. Samples mounted directly on stub have been simply chrome-

coated. SEM images have been taken with a Field Emission Gun Scanning Electron Microscope (SEM-FEG) Gemini 500 Zeiss, and a Tescan VEGA TS 5136XM. Standard magnification for SEM images were established (~2500X, ~5000X, ~10000X, ~20000X and ~30000X ), to describe comparatively and measure growth bands and cells, the morphology of Mg-calcite crystals, and the main features of perithallial and epithallial cell walls. A rigorous control over cell orientation is required to represent, describe, and measure in 2D the main features of a 3D structure such as cell calcification (both PW and SW).

Longitudinal axial sections of branches are a standard representation for calcareous red algae, allowing for subsequent visual comparison (Woelkerling, 1988; Quaranta et al., 2007; Burdett et al., 2011). Surface tangential sections are useful to describe the epithallial cells. Transverse or oblique sections are useful to describe qualitatively the three-dimensional aspects and organization of calcite crystals composing both PW and SW. Description of the cell wall structure follows the nomenclature of Flajs (1977), presenting the primary (PW) and secondary (SW) calcifications of the wall. Some authors refer to PW as

interstitial calcite (Ragazzola et al., 2016) or interfilament calcite (Nash and Adey, 2017; Nash et al., 2013, 2015). Cell dimensions have been measured as reported in Fig. 2, exclusively on longitudinal sections (Fig. 2). Separation among adjacent filaments was not always obvious (Fig. 2c). In such cases, PW of adjacent cells has been measured in total (green line in Fig. 2c) and then half of the total was attributed to each cell.

**2.3 Statistical analyses**

Spearman's and Pearson's correlations were used to test the statistical relationship between the cell measurements in both long and short cells, including morphometry and cell wall thickness. The linear correlation between the mean cell lengths and the sampling depths was measured by Pearson's coefficient, as well. One-way ANOVA and the Kruskal-Wallis test respectively followed by the Tukey's test and the Dunn's test for post-hoc analysis was used to compare the cell measurements among sampling sites and to evidence the differences between group means and medians. All statistical analyses were run in R 3.6.3

software.

**3. Results**

**3.1 Ultrastructures from SEM images**

Selected rhodoliths belong to the unattached branches morphotype (Basso, 1998; Basso et al., 2016), never exceeding 3 cm in length (Fig. 1). The diameter of each branch never exceeds 2.5 mm (Fig. 1). The color varies from yellowish white to

pink/purplish, with typical knob-like apices (Fig. 1).

Once cut, all samples show the same micromorphology (Fig. 3a-i), with the constant occurrence of an easily detectable banding due to the alternation of series of short and long cells (Fig. 3a, b). No reproductive structure (conceptacle) was detected.



Along the perithallus, long cells are ovoid to rectangular in shape (Fig. 3c, d), whereas short cells are more squared (Fig. 3e, f).

Within the long-celled bands, the longest cells were measured in the sample from Morlaix Bay (26.71±1.74 µm), while the less elongated cells were observed in the sample from the Pontine Islands (13.05±0.76 µm) (Table 2, Fig. 4).

Within the short-celled bands, the shortest cells were observed in the sample from Pontine (6.97±0.25 µm), while the longest short cells were from Morlaix Bay (13.90±0.88 µm) (Table 2, fig. 4). Cell diameter ranges between 7.69±1.07 µm (Santa Caterina shoal) and 11.11±1.79 µm (Morlaix) in long cells, and between 8.27±0.48 µm (Pontine) and 9.23±0.70 µm (Egadi)

in short cells (Table 2, Fig. 4).

Both long and short cells have PW and SW walls, and the style of mineralization shows a consistent ultrastructural pattern. The PW crystallization observed in longitudinal medial sections (cutting the cell lumen) is composed of elongated crystals appearing as rods (Fig 5a-c, e), but where longitudinal sections are tangential to the PW, crystals reveal to be flat rectangular tiles (Figs. 5d, f, 6a). The long axis of the PW tiles is parallel to the cell membrane and may form a multi-layered structure,

which envelops the cell (Fig. 5e). The dimensions of the tiles are 0.61±0.17 µm in length and 0.05±0.01 µm in height (Fig. 6a). Elongated radial crystals (Fig. 7a, b) in longitudinal sections cutting the cell lumen characterize the SW. Small, roundish crystals appearing as fused together, in places showing an apparent multi-layered structure (Fig. 7b) form such crystals. These elongated crystals are radial to the cell lumen (Fig. 7a-c). Where the cell membrane is lost, SW can be observed from different orientations, and such apparently elongated crystals reveal to be thin bricks with rounded margins (Figs. 6b, 7d-f). Bricks are

squared, and length and width are 0.63±0.15 µm (Fig. 6b). The bricks form a sort of envelope around the cell (Figs. 6b, 7e, f) showing sometimes a zigzag pattern (Fig. 7g, h).

Different thicknesses characterize PW and SW of short and long cells (Figs. 3, 5, 7; Table 2) in longitudinal medial sections. Both PW and SW of short cells are generally thicker than in long cells (Table 2), even if the thickness does not show a correlation with sampling bathymetry. SW thickness ranges between 1.52±0.65 µm (Pontine) and 2.20±0.45 µm (Elba) in

short cells, and between 0.57±0.14 µm (Morlaix) and 1.26±0.42 µm (Santa Caterina shoal) in long cells (Table 2).

Both PW and SW show fibrils (Borowitzka, 1982) forming a dense network in support of the mineralization (Fig. 7d).

Epithallus is formed by one up to three flared cells in longitudinal sections (Figs. 3g-j, 5f, 7c), always mineralized, with some exceptions in the top distal surface (Fig. 3 g-j). The cell wall shows the same ultrastructural features of the perithallial cells, with both PW and SW mineralized (Figs. 3i, j; 5f, 7c).

Basing upon image analysis and time of collection, the calculated growth-rate ranges from 0.10 mm/yr in Pontine to 0.13 mm/yr in both Morlaix and Egadi.

The two additional collections, *L. valens* and *L. minervae* (Fig. 8), are characterized by both PW and SW (Fig. 8a, b, e, f). This last shows, in both cases, an ultrastructural arrangement similar to the one described for *L. corallioides* collections, with differently oriented bricks with rounded margins only apparently elongated and radial to the cell lumen in longitudinal sections

(Fig. 8b, f). If observed into the cell lumen, where the cell membrane is lost, SW shows bricks with different orientation and sometimes a zigzag pattern (Fig. 8d, g).





On the contrary, PW shows a different shape and arrangement of crystals, which are not characterized by the tiles of *L. corallioides* observed in Figures 5d and 6a. Calcite crystals are more squat and granular (Fig. 8a) or with irregular shape (Fig. 8h). One interesting aspect is that both samples show the occurrence of secondary calcite in form of dogtooth crystals filling

the cell lumen (Fig. 8b, c, e).

**3.2 Statistical analyses on morphological parameters**

The differences in the long cells' morphometry and wall thickness among sampling sites are statistically significant for each measured parameter ($p<0.05$; Supplement 1) (Fig. 9). Interestingly, the long-cell length of the deepest sample from Pontine Isl. (66 m depth) is lower than the others ($p<0.001$) (Figs. 4, 9; Supplement), while in the shallowest sample collected in

Morlaix (12 m depth) cells are notably longer ($p<0.001$) (Figs. 4, 9; Supplement 1).

In short cells, remarkable differences result only for cell (Fig. 9) and lumen lengths, and cell PW ($p<0.05$; Supplement). The shortest cells are observed again in the sample from Pontine Isl., differing from the one collected in Morlaix ($p<0.01$) (Fig. 9; Supplement 1), which outstands for the highest values. On the contrary, the cell diameter slightly varies among sites, showing significant differences just in long cells ($p<0.01$; Supplement 1) (Fig. 9), with significantly higher values in Morlaix with

respect to Elba Island and Santa Caterina. The dimensions of the cell lumen change accordingly, because of the positive correlation with the cell dimensions (Supplement 1).

Although banding is reported for all samples, elongation decreases with increasing depth, showing a strong inverse correlation in long cells ($p<0.01$; $r=0.98$) (Fig. 9). The same trend is also observed in short cells, even with non-significant values ($p=0.09$) (Fig. 9).

**4 Discussion**

Properly oriented longitudinal and transverse/oblique sections are mandatory to obtain a precise comparative description of the main morphological features of CCA. Multi-scale approaches are also relevant, among which the ultrastructural pattern may represent a new powerful and strategic diagnostic tool (Figs. 3-7).

Based upon macroscopic features, thallus pattern, microanatomy, morphology, and morphometry of cell walls (Figs. 3, 5-7,

Table 2) we identified the samples from Morlaix Bay as belonging to the species *L. corallioides*. Recent studies based upon genetic identification exclude the occurrence of other *Lithothamnion* species in the maerl of Morlaix Bay (Carro et al. 2014; Melbourne et al., 2017). Therefore, we compared the Mediterranean specimens with the Atlantic *L. corallioides*, and upon corresponding morphology, anatomy, and ultrastructure (Figs. 3, 5-7, Table 2), we considered them as conspecific.

The perithallus of *L. corallioides* clearly shows the alternation of growth band of third and fourth orders (Fig. 3a, b), in

agreement with Foster (2001). Fourth order bands represent the annual cycling, whereas third order ones represent seasonal variations and can be firstly distinguished by an evident chromatic change due to the different calcification thickness between long and short cells (Foster, 2001). In our samples, banding (third and fourth orders) is easily recognizable (Fig. 3), and both long and short cells length decrease across depth (Figs. 4, 9), as expected, mirroring a decrease in growth-rate, whereas diameter variation is significatively lower (Fig. 4, Supplement 1).



Giraud and Cabioch (1979) observed that a cell wall fracture in *L. corallioides* shows a layer of radial calcite crystals (SW) separated from its neighbor by a different sheet composed of tangential crystals (PW). A discontinuity that coincides with the fibrillar matrix observed in sections of decalcified material marks the limits of adjacent cellular frames (Giraud and Cabioch, 1979). Our results match with the observation of these authors in longitudinal sections (Figs. 3, 5, 7), although the discontinuity between adjacent cells is not easily detectable because of the total mineralization.

Auer and Piller (2020) built a morphological tree based upon the observation of different ultrastructural patterns in epithallial cells, which match with the phylogenetic tree at family level. For Hapalidiaceae, and in the Lithothamnion-type epithallial ultrastructure, they observed the occurrence of PW with primary crystals formed along the middle lamella and the SW with secondary rod-shaped crystals, also presenting fan-like structures. The samples studied in the present work show PW and SW both in perithallial and epithallial cell walls (Figs. 3, 5, 7). Both are composed of apparently rod-like crystals in longitudinal

sections. However, on longitudinal sections that are locally tangential to the PW, the apparent rods reveal to be the longest and thinnest side of variably oriented rectangular tiles (Figs. 5, 6a). The tiles that envelop the cell (Fig. 5d, f) are the basic ultrastructural elements forming the PW. Differently, apparent rods of the SW reveal to be squared and relatively flat bricks with rounded margins as observed at the cell lumen without membrane or exactly at the contact between SW longitudinal section and cell lumen (Figs. 6b, 7). Crystals in longitudinal sections of SW are radial to the cell lumen, in agreement with

Giraud and Cabioch (1979), and seem formed by small grains fused together (Figs. 3, 7b, f). They can also show a zigzag pattern (Fig. 7g, h) like the fan-delta structure described by Auer and Piller (2020).

Therefore, our findings agree with the results of Auer and Piller (2020), although providing a more precise description of the ultrastructural pattern of *L. corallioides* (Fig. 6).

Despite the different environmental conditions, likely occurring at the different sampling sites and depths, the ultrastructures

of both PW and SW seem conservative and detectable in all samples. Therefore, the ultrastructural pattern is not-dependent on environmental controls. However, *L. corallioides* shows variable cell elongations (Table 2) and growth-rates, decreasing both with according to sampling depth (Fig. 9), and a variation in PW and SW thickness, generally major in short than in long cells, not-dependent on depth (Table 2, Supplement 1).

These features possibly represent the effect of the different environmental conditions in which it lives, but without affecting

the ultrastructural pattern. These findings support the idea that the calcification process in CCA is biologically-controlled (Borowitzka, 1984; Cabioch and Giraud, 1986), rather than induced (de Cervalho et al., 2017; Nash et al., 2019).

The two additional samples, L. minervae and L. valens, show distinct styles of PW calcification, and this is extremely interesting for its application in Paleontology. Further investigation is needed to clarify the validity of this hypothesis in other genera/species.

Finally, the occurrence of secondary calcite, in form of dogtooth crystals that fill the cell lumen (Fig. 8b, c, e), is a novelty. The voids of the cell lumens allowed the development of calcite crystals which in term of shape, dimension and pattern are completely different from the ones forming the cell wall calcification. It represents a form of extremely early diagenesis in collections that were alive at time of sampling. The phenomenon of early diagenesis has implications in the reliability of



climate and paleoclimate studies based on the geochemistry techniques, also when applied to relative recent collections, as
already indicated for Holocene Scleractinia corals (Nothdurft and Webb, 2008; Rachid et al., 2020). Therefore, this
phenomenon should be considered very carefully also for coralline algae when selecting samples for this type of studies.

**5. Conclusions**

We define the cell-wall ultrastructural pattern of *L. corallioides* as follows:

- perithallus with evident banding as the result of the alternation of series of short-squared and long ovoid/rectangular cells;
- epithallus with one up to three flared cells;

- same and consistent ultrastructural pattern of the cell walls both in perithallus and epithallus, with PW and SW calcite walls
always present;

- the PW is characterized by rectangular tiles;

- the SW is characterized by flattened squared bricks with roundish outlines;
- long and short cells have similar diameter, with different thickness of PW and SW, resulting mainly in a thicker PW and SW
in short cells.

The variable cell elongation, decreasing according to depth, and producing an evident banding, never affects the ultrastructural
pattern that maintains the same arrangement also under different growth-rates. Therefore, the calcification process of CCA
seems to be biologically-controlled rather than induced. The comparison with other *Lithothamnion* species highlights
difference in the mineralization pattern of PW. Therefore, the ultrastructures of the cell wall in CCA results to be a promising
new diagnostic tool for species identification with important potential application in Paleontology. Lastly, early diagenesis
phenomenon, at the scale of ultrastructures, have been identified for the first time in living coralline algae.

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

Bracchi Valentina Alice Bracchi conceived the research, conducted SEM observations, interpreted the results, and wrote the manuscript.

Piazza Giulia Piazza conducted the statistical analyses.

Basso Daniela Basso conceived the research and wrote the manuscript.



**Captions, Figures and Tables**

Figures

Figure 1: Sampling sites (1-5) and images of selected samples. Service Layer Credits: Sources: Esri, GEBCO, NOAA, National Geographic, Garmin, HERE, Geonames.org, and other contributors Esri, Garmin, GEBCO, NOAA NGDC, and other contributors. Scale bar = 1 cm.

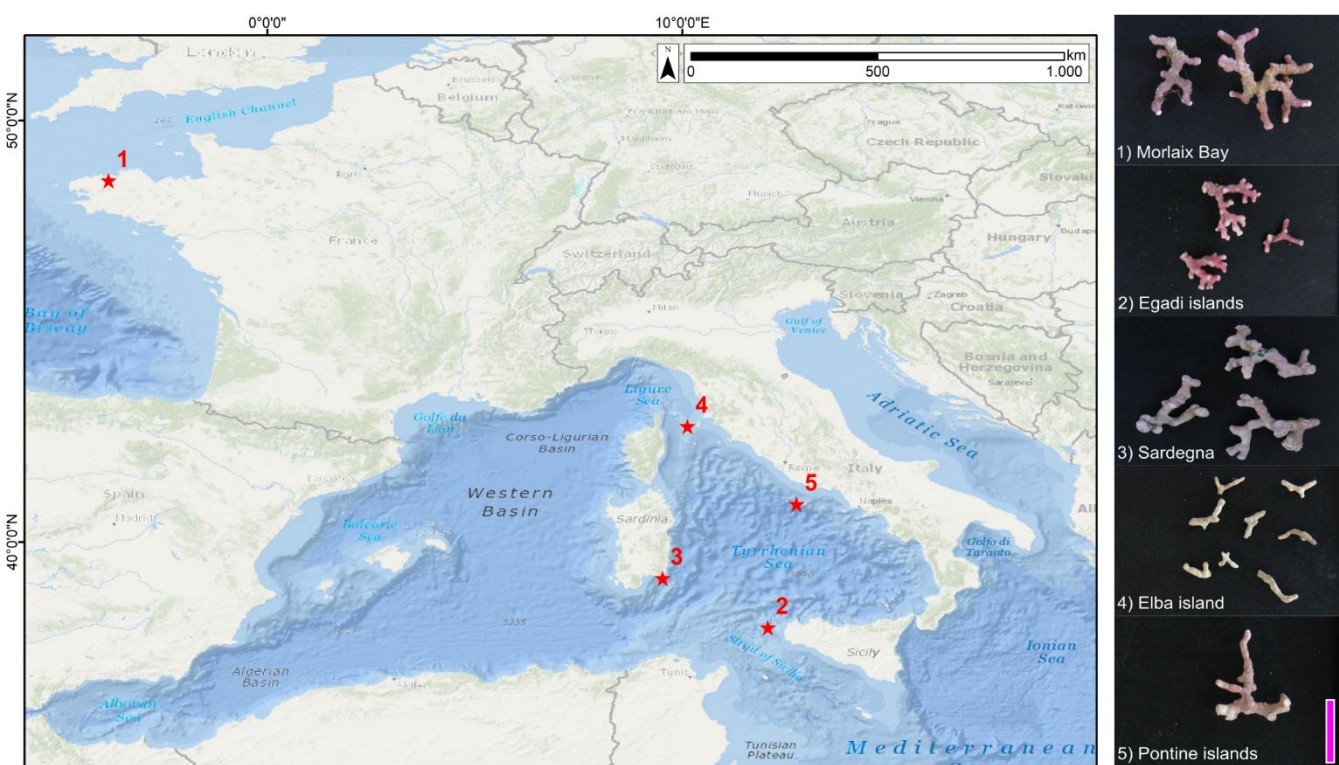






Figure 2: Cell measurement: a) cell wall length (red) and diameter (blue); b) lumen length (yellow) and lumen diameter (purple); c) SW thickness of adjacent cells (orange); PW thickness of adjacent cells (light blue).

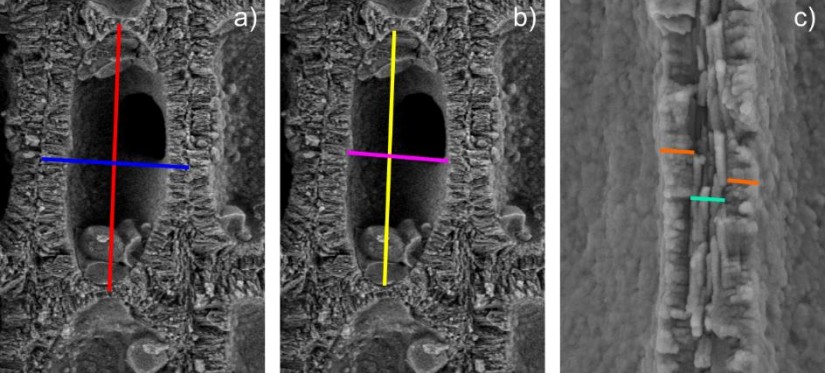


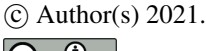



Figure 3: Main features of Lithothamnion corallioides under SEM (Morlaix Bay, France):

a) longitudinal section of a fragment of L. corallioides with obvious banding (black dashed lines). Scale bar = 100 µm. The inset is magnified in b). Scale bar = 100 µm;

b) alternation of thick-walled (black arrow) and thin walled (white arrow) cells. The inset is magnified in c). Scale bar = 20 µm;

c) Thin-walled long cells. The inset is magnified in d). Scale bar = 10 µm;

d) Detail of the thin wall of long cells. Scale bar = 1 µm;

e) thick-walled short cells. The inset is magnified in f). Scale bar = 2 µm;

f) detail of the thick-wall of short cells. Scale bar = 1 µm;

g) polygonal shape of epithallial cells in surface view. Scale bar = 10 µm;

h) detail of a flared epithallial cell (arrow). Scale bar = 3 µm;

i-j) longitudinal sections of flared epithallial cells with complete mineralization of the cell walls. Scale bar = 2 µm.






Figure 4: Box plots reporting cell lengths and cell diameters in both long and short cells of L. corallioides samples collected at different sampling sites, ordered along x-axis according to the depth.





Figure 5: Details on primary wall (PW) in longitudinal section:

a) thick-walled short cell showing both PW (black arrow) and secondary wall (SW, white arrow), Santa Caterina shoal. Scale

bar = 2 µm;

b) thick-walled short cell showing both PW (black arrow) and SW (white arrow), Pontine Islands. Scale bar = 2 µm;

c) thin-walled long cell showing both PW (black arrows) and SW (white arrow). PW crystals in longitudinal section appear as

elongated crystals, Santa Caterina shoal. Scale bar = 1 µm;

d) thin-walled long cell showing both (black arrow) and SW (white arrow). The fracture shows detail of tiles composing the

PW (black arrow), Santa Caterina shoal, scale bar = 1 µm;

e) details of multi-layered PW (black arrow) and SW (white arrows). PW crystals in longitudinal section appear as elongated

crystals. Pontine Islands, scale bar = 0.2 µm;

f) perithallial and epithallial cells where the longitudinal section is locally tangential to the PW (black arrows): crystals appear

as flat rectangular tiles, Santa Caterina shoal. Scale bar = 2 µm.

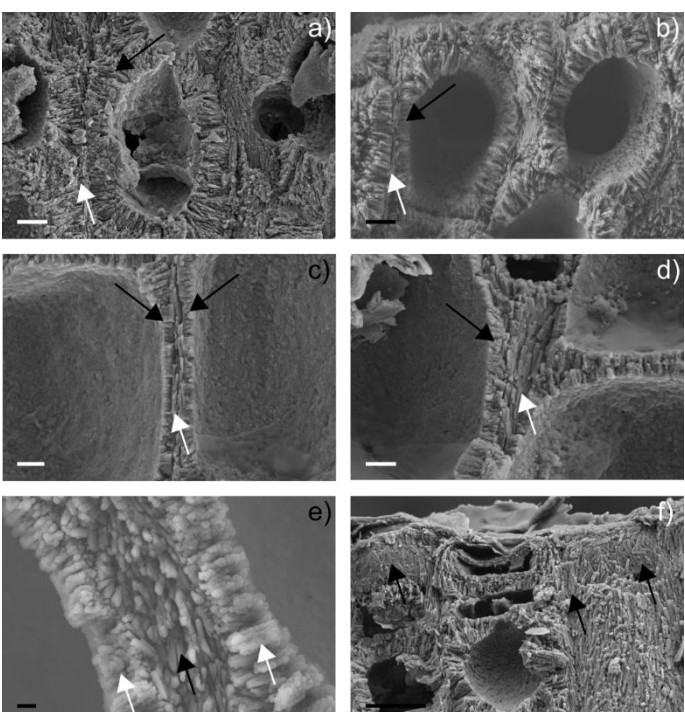





Figure 6: Ultrastructures of Lithothamnion corallioides: a) Rectangular tiles of the PW; b) squared bricks of SW.

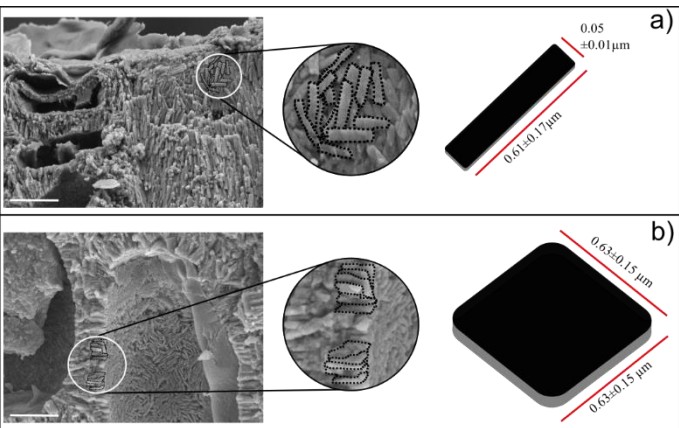






Figure 7: Details on secondary wall (SW) in longitudinal section:

a) thick-walled short cells SW showing both PW and SW, Morlaix Bay. Inset magnification in b). Scale bar = 2 µm;

b) Two adjacent cells with both PW and SW. The secondary wall is characterized by elongated radial crystals formed by small

roundish crystals appearing as fused together, in places showing an apparent multi-layered structure. Note the occurrence of a very thin PW (black arrow). Morlaix Bay. Scale bar = 1 µm;

c) perithallial thick-walled short cells showing both PW and SW, and one-layer epithallial cell with the evidence of a flattened cell (red arrow), Morlaix Bay. Inset 1 magnification in d), inset 2 magnification in e) and inset 3 magnification in f). Scale bar = 2 µm;

d) magnification of inset 1 in c) showing SW in lumen cell with no membrane. In this view the crystals appear as ovoidal to rod-shaped with a complex orientation, associated with fibrils. Morlaix Bay. Scale bar = 1µm;

e) magnification of inset 2 in c) showing SW in lumen cell with no membrane. A section showing at the same time a longitudinal section and the inner cell reveal the actual 3D shape of the SW crystals that are thin bricks (white arrows). Morlaix Bay, scale bar = 1 µm;

f) magnification of inset 3 in c) showing SW in lumen cell with no membrane. A section showing at the same time a longitudinal section and the inner cell reveal the actual 3D shape of the SW crystals that are thin bricks (white arrows). Morlaix Bay, scale bar = 1 µm;

g) a detail of SW inside a cell lumen. The crystals are thin bricks with a complex zig-zag orientation. Inset magnification in h). Egadi Islands, scale bar = 2 µm;

h) a zigzag and/or crossing pattern of bricks in SW. Egadi Islands, scale bar = 1 µm.



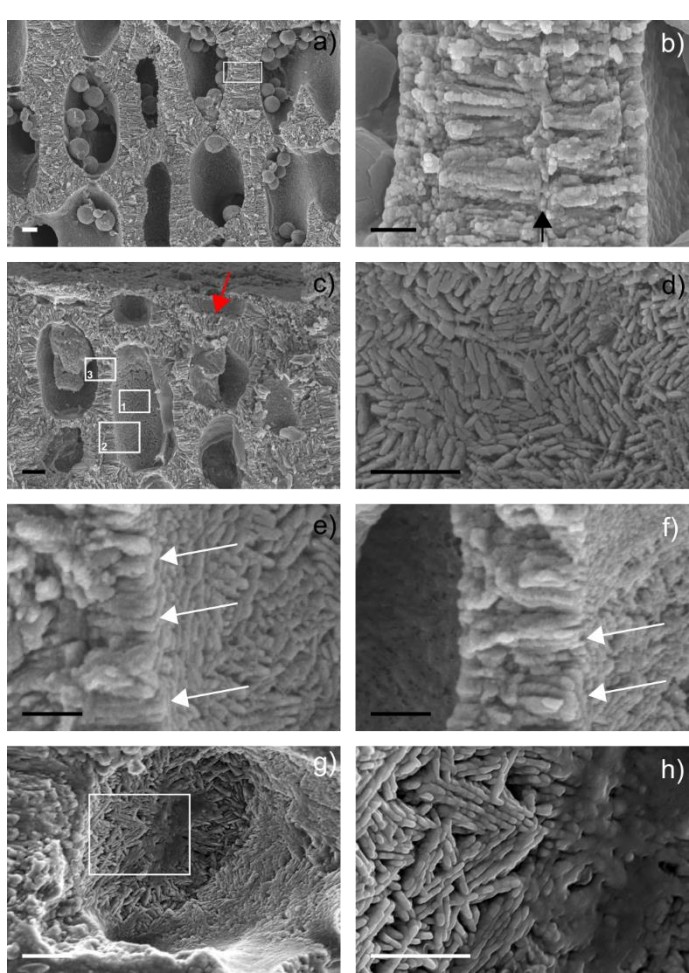





Figure 8: Details on the main ultrastructures of Lithothamnion valens (a-d) and Lithothamnion minervae (e-h):

a) perithallial and epithallial cells showing both PW and SW. Scale bar = 2 µm;

b) details of perithallial cell with both PW (black arrow) and SW (white arrow). The secondary wall is characterized by elongated radial crystals, whereas PW crystals are tangential to the cell lumen. The cell lumen is filled by secondary calcite (dogtooth shape, red arrows). Scale bar = 1 µm;

c) detail of a cell wall with evident PW (black arrow) and SW (white arrow), and secondary dogtooth calcite filling the cell lumen (red arrow). Scale bar = 2 µm;

d) detail of SW (white arrow) into the cell lumen. Note the multi-direction arrangement of calcite crystals. Scale bar = 2 µm;

e) perithallial cell, with elongated and rectangular shape. The central cell shows a cell lumen filled by secondary calcite with dogtooth shape (red arrow). Scale bar = 2 µm;

f) perithallial cell with both PW (black arrow) and SW (white arrows). Note in the cell lumen that SW is characterized by elongated crystals showing different orientation and fan-delta arrangement. Scale bar = 2 µm;

g) detail of f, with the SW into the cell lumen, with elongated crystal showing different orientation and fan-delta arrangement. Scale bar = 1 µm;

h) the fracture shows detail of crystals composing the PW (black arrow) apparently forming by granules. SW is characterized by elongated crystals in longitudinal section (white arrow). Scale bar = 1 µm;








Figure 9: Correlation plots showing the relationship between sampling depth and cell lengths, measured in both long and short cells. Pearson's correlation significance at p<0.05. Pink is for Morlaix Bay (France, 12); green is for Egadi islands (Italy, 40); red is for Santa Caterina shoal (Italy, 40); yellow is for Elba island (Italy, 45); Blue is for Pontine islands (Italy, 66).

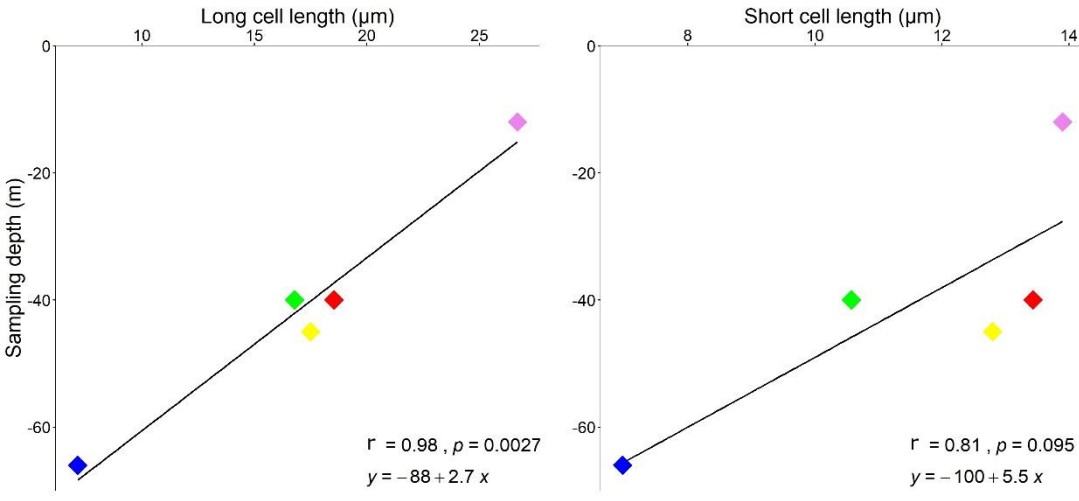






Tables

Table 1: Location, date of collection and depth of samples. In Basin column, numbers in brackets corresponds to the point in Figure 1.

| Basin | Sample | Location | Date of collection | Depth (m) |
|---|---|---|---|---|
| Atlantic Ocean (1) | France, Morlaix Bay | 48°34′42″N 3°49′36″W | May, 1991 | 12 |
| Western Mediterranean (2) | Italy, Egadi Islands | 37°58′10″N 12°03′26″E | August, 1991 | 40 |
| Western Mediterranean (3) | Italy, Santa Caterina shoal | 39°08′32″N 9°31′14″E | July, 2017 | 40 |
| Western Mediterranean (4) | Italy, Elba island | 42°44′56.4″N 10°07′08.4″E | December, 1990 | 45 |
| Western Mediterranean (5) | Italy, Pontine islands | 40°54'N 12°45'E | July, 2016 | 66 |




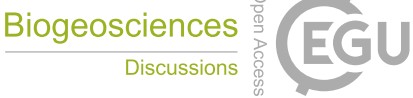

Table 2: Morphometry of short and long cells/lumens and wall thickness measured in longitudinal section (PW = primary wall, SW = secondary wall), with the indication of length (L) and diameter (D) (μm). Standard deviation in brackets.

| Sample | Short cell | | | | | | Long cell | | | | | |
| | Cell | | Lumen | | Wall | | Cell | | Lumen | | Wall | |
| | L | D | L | D | SW | PW | L | D | L | D | SW | PW |
|---|---|---|---|---|---|---|---|---|---|---|---|---|
| Morlaix Bay 12 m | 13.90 | 8.31 | 12.38 | 6.04 | 1.61 | 0.12 | 26.70 | 11.11 | 24.46 | 9.25 | 0.57 | 0.13 |
| | (0.88) | (1.34) | (0.93) | (1.05) | (0.43) | (0.03) | (1.73) | (1.80) | (1.73) | (1.42) | (0.14) | (0.06) |
| Egadi islands 40 m | 13.43 | 9.22 | 8.48 | 4.97 | 2.08 | 0.22 | 18.55 | 10.27 | 16.13 | 8.54 | 0.74 | 0.16 |
| | (2.36) | (0.70) | (1.189) | (1.55) | (0.80) | (0.07) | (1.28) | (0.77) | (1.16) | (0.60) | (0.09) | (0.05) |
| Santa Cat. shoal 40 m | 10.57 | 8.29 | 8.79 | 4.43 | 1.87 | 0.17 | 16.80 | 7.69 | 15.40 | 5.04 | 1.25 | 0.13 |
| | (1.01) | (0,58) | (0.72) | (0.66) | (0.26) | (0.16) | (1.54) | (1.07) | (1.51) | (0.81) | (0.41) | (0.03) |
| Elba island 45 m | 12.8 | 9.4 | 11.6 | 5.2 | 2.2 | 0.14 | 17.51 | 8.64 | 15.75 | 6.31 | 1.08 | 0.18 |
| | (0.45) | (0.9) | (0.54) | (0.44) | (0.44) | (0.05) | (0.78) | (0.81) | (0.89) | (0.83) | (0.25) | (0.06) |
| Pontine islands 66 m | 6.97 | 8.27 | 5.19 | 4.49 | 1.79 | 0.47 | 13.05 | 9.64 | 10.38 | 6.81 | 1.22 | 0.33 |
| | (0.25) | (0.47) | (0.72) | (0.31) | (0.27) | (0.08) | (0.76) | (0.83) | (0.54) | (0.87) | (0.17) | (0.09) |


