# Peer review of "A stable ultrastructural pattern despite variable cell size in"

_Biogeosciences, 2021_

## Author Response (AR2)

RC1: Responses to the comments of Reviewer 1 in blue:

The paper contains a detailed and well-illustrated description of the structure of calcified cell walls in specimens of *L. corallioides* from different settings. The paper shows that skeletal ultrastructure of L. corallioides does not change in environments substantially different in terms of illumination (depth), temperature, and salinity. Based on the maintenance of the same ultrastructural patterns in different environmental conditions the authors conclude that 'the calcification process of CCA seems to be biologically-controlled rather than induced'. As suggested in the introduction, the aim of the paper is to contribute to the debate about the nature of calcification in coralline algae. Its results support that calcification is biologically controlled and, therefore, refute the conclusion of Nash et al. (2019) that mineral formation in corallines is biologically induced. The latter authors, however, based their interpretation on a detailed discussion of several features of calcification of corallines and any rebuttal of their conclusion should address the same features on the light of the new findings. I believe that a discussion of features that according to Nash et al. (2019) are key to decide whether the mineral formation is induced or controlled must be included in the paper.

Dear reviewer, thank you for your revision.

The study of coralline algae represents in general a challenge for scientist, and the mechanisms of calcification are far from been understood.
The paper of Nash et al. (2019) hypothesized a time-step scheme of biomineralization based on the observation of a wide dataset (24 species) which considers both crustose coralline algae (non-geniculate) and articulated (geniculate) corallines. In the light of their observations, they not only hypothesize such model, but also support that the biomineralization in coralline algae is a biologically induced process basing upon a series of key-assumptions.

The biomineralization is the process of formation of a mineral phase carried out by organisms. It depends on the degree of biological control over the formation (Lowenstam, 1981; Weiner and Dove, 2003, Päßler et al 2018) ranging between two mechanism: the biologically controlled mineralization (BCM), in which organisms have extensive control over the mineral formation, or biologically induced mineralization (BIM), in which organisms have no to minor control over the mineral formation. BCM results in well-ordered mineral structures, with minor size variations and species-specific crystal habits (Bazylinski and Frankel, 2003), whereas BIM results in heterogeneous mineral compositions with poor crystallinity, including large size variations, poorly defined crystal morphologies and the inclusion of impurities (Banfield and Hamers, 1997; Frankel and Bazylinski, 2003; Weiner and Dove, 2003).

Our study is aimed at describing, in a standardize way (longitudinal section), the main ultrastructural features, which represents the result of biomineralization, of cell wall in *L. corallioides.* Having collected the same species but from difference geographic context and water depths, we also test if this translates in ultrastructural differences at level of calcite crystals in the cell wall. Basing upon our observation, we proposed an ultrastructural pattern for *L. corallioides.* In the light of our results referring to ultrastructures (well-ordered, minor size variations, species-specific crystal habit), we support that the biomineralization in coralline algae is biologically controlled rather than induced, perfectly adhering the original definition.

In the case of Nash et al. (2019) layers of the cell wall (and not ultrastructures) and their mutual organization are described using figures in which algae are apparently randomly oriented and this

is the first problem in comparing our results. Nevertheless, we found some similarities between our results and theirs, in term of layers forming the cell walls (primary, secondary), thickness of cell wall, structure of grains, but also substantial differences (not only PCW cell). These results, similarities and differences, referred to morphological units characterizing the cell wall, will be more discussed in the revised version of the manuscript. It was not our scope to propose another hypothetic scheme of biomineralization, nor rebut to all the assumptions the authors did in their paper, but we will insert some specific rebuttals, which concern only the mineralogical aspects in the revised version of the manuscript.

Minor points: Check species names are in italics. We will check it carefully.

RC2: Response to the comments of Reviewer 2 in blue:

General Comments
In this paper, the microstructure of the chosen coralline algae was carefully observed at each depth using electron microscopy. The study can make a significant contribution to the community that is studying the diversity of coralline alga. It is important to note that the patterned microstructure in this species is also helpful for identifying these algae. The suggestion that there may be a correlation between water depth and cell size could also enhance the value of this study.
However, the fact that the number of individuals used for observation and measurement has not been clarified is an essential deficiency when considering the reproducibility of this study. The number of individuals in the samples used should be clearly stated.
We will review the manuscript to clarify this aspect and add all the details on how many samples we consider for our study.

In addition, to refer to the "very early diagenesis," it is essential to describe the more detailed processing and storage conditions from sample collection to observation. We believe that the current description does not go as far as to say that diagenesis effects were found in living individuals.
Thank you for this comment. We will include all the details on storage and processing you requested in the reviewed version of the manuscript. We will be more conservative in defining our observation.

The authors should also concern about the lack of environmental data other than the water depth. Currently, they are discussing the correlation with water depth, but various parameters such as light intensity, water pressure, water temperature, and others co-vary with water depth. To examine which of these parameters has more influence, environmental data other than water depth needs to be presented.
This is a good point, and we agree with you that parameters such as light intensity, water pressure, water temperature, and others co-vary along with the water depth gradient and affect the development of coralline algae. We will add the values (temperature) of such variables on the base of regional databases.
We here refer to the sampling depths to indicate the possible differences that we expected to have in their life environments. The aim of this study is to see if the same species, collected from different geographic context and depths, can modify its mineralogical structures in the wall because of differences in the living context.

Moreover, it would be better to observe the micromorphology from more angles to compose diagrams for comparison. That diagrams can be helpful for general comparison among other groups of organisms and others. The crystal photographs from various angles would make the crystal's morphology easier to understand for a general audience.
Thank you for this comment. On one hand, we agree that the observation from more angles allow to compose a better view of the crystals and we already used pictures from different angles (Fig. 3g, 5b, d, f, 7e, h). On the other hand, only a standardize point of view (longitudinal cut of the branch) allows the right description of the morphometry and shape of crystals as defined in Figure 6.
We would be grateful to have further indications about the type of possible additional diagrams.

**Individual comments**
P2L34 Why does examining GCA reveal the primary production of macroalgae? Is it correlated with the local/regional overall macroalgae?

This sentence refers to the fact that coralline algae, being calcareous, are among macroalgae the most important in the fossil record, thus providing at least a window about the primary production of the past

P3L77 This reviewer is not sure of the motivation that led to this objective. The authors' working hypothesis is that the elements depend on environmental factors, or are they stable within the region? If the motivations of this study are not clear, it is not possible to judge whether this study is appropriate for BG audiences.
The aim of this study, as declared at the end of the Introduction paragraph, is to describe in a standardized way the ultrastructures of cell wall in *Lithothamnion corallioides*, which was collected in different geographical context and depth, and to test if there is a morphological and morphometrical variations in such structures which can be related to environments in which it lives or if they are unaffected by environments. This is the first contribution to the knowledge of the ultrastructures in *Lithothamnion corallioides* and the "behavior" of these crystal elements under different environmental conditions.

This reviewer is not confident that the authors have drawn appropriate conclusions (P8L237) that will be of interest to the BG audience. Perhaps it would be better to publish the paper as a classification study in a more paleontological journal to reach the right audience.
We proposed our study to BG because this journal (as stated in the webpage) is "dedicated to all aspects of the interactions between the biological, chemical, and physical processes in terrestrial or extraterrestrial life with the geosphere, hydrosphere, and atmosphere". The biomineralization process, not only in the coralline algae, is exactly one of the natural processes that involves interactions among biological, chemical, and mineralogical aspects. With our work, we highlighted that from the mineralogical point of view we found out that the ultrastructures are stable. The fact that our results can be of interest also for paleontologist represents an addendum to our results.

Further, the authors should clarify what elements are included in the" ultrastructural mineralogical features" and "ultrastructure pattern."
In the text, we used both definitions depending if we referred to the singular crystal elements (ultrastructural mineralogical feature) or to the combination and mutual organization of crystal elements forming the entire wall (ultrastructure pattern). If this led to misunderstanding, we will add the definition in the Material and Methods paragraph in the reviewed version of the manuscript.

P3L86 Is the identification of samples done by molecular biology? Since authors are discussing variation in microstructural morphology, would not species identification by morphology be a circular argument?
This is already explained in lines 190-195. Recent studies based upon genetic identification exclude the occurrence of other *Lithothamnion* species in the maerl of Morlaix Bay (Carro et al. 2014; Melbourne et al., 2017). We collected samples exactly in the same locality and we attributed them to *L. corallioides*. On this basis, we identified the ultrastructures of *L. corallioides* (Figure 3). Then we compare our results with the other sample (Figure 5, 7). Therefore, our argument is not circular.

P3L87 The authors discuss the very early diagenesis of crystals. It is crucial to describe the details of the method from collection to sample preparation to ensure reproducibility. Please describe it. It is acceptable to make supplemental material.

Thank you for the suggestion. We will include all the missing details on storage and transport in the reviewed version of the manuscript. For what it may concern the processing of samples, we already inserted all the details in the M&M paragraph.

Readers need to know the environmental information of the sample collection point. For example, in Fig.4, we can consider whether the depth or environmental information such as temperature is correlated with the morphological features.

As already stated before, we used the same species (*Lithothamnion corallioides*), collected in different geographical context and depth (that means for sure different environmental conditions), to test if there is a morphological and morphometrical variations in ultrastructures of the cell wall possibly related to such variation. Our results clearly show that ultrastructures are unaffected by environmental conditions. Nevertheless, we will add the range of temperature at which the collected samples lived (last years before collection) on the base of regional databases (Table 1), which is primarily indicated as responsible of the banding in CCA, to give reader an idea of seasonal variation in term of temperature.

P3L95 Could the authors list the number of samples for each branch at each depth in the text or Table 1.

Ok, we will add it in the revised version of the manuscript (Table 1).

P4L116 The number of cells measured and other information for each sample should be indicated. The authors can also write the number of samples (n=) in Figure 4 or Table 2.

Ok, we will add it in the revised version of the manuscript (Table 1).

P4L127 To enable the reader to easily compare differences between samples, a picture of the long and short cells at all water depths should be shown in a single figure for comparison.

We have already 8 figures in the paper. Therefore, we will prepare it as Supplementary 2 material it in the revised version of the manuscript.

P4L131 If the combination of short and long cells is the seasonal variation, please show the reason for the classification into two types: short and long. If it is a seasonal variation, some may have a transitional feature. For example, if authors make a scatter plot, it may divide into two. Another possibility is that the cells appear to be short due to the difference in cutting direction. To prove this, MXCT measurement is recommended to show that the cells are smaller in three dimensions.

The interpretation of banding as the results of the alternation of long and short cells due to seasonal alternation is an obvious feature that is visible also at low magnification, and a widely observed and accepted pattern in the structure of coralline algae thalli (Foster, 2001 and references herein). One band is easily identifiable because of the occurrence of long cells at the base, passing to short cells at the top. Our observations were conducted on carefully checked longitudinal sections in order to avoid error of cutting direction. This can be checked by the identification of primary pit connections connecting cells of the same filaments.

If short cells and long cells seem to have different cytological roles, to begin with, that explanations should be introduced beforehand in the introduction.

We do not declare any different cytological roles between short and long cells. We just indicated that there are differences in the dimension of the cell in the same samples.

P4L144 Are "flat rectangular tiles" the form of crystal defined in previous studies? If there is already terminology for similar forms in previous studies, please cite it to avoid future confusion. Also, this reviewer is not convinced that the name "tiles" and the schematic diagram are appropriate for the crystal form seen in this picture. It would be good if SEM pictures from various angles could be available.

There are no previous studies on the ultrastructures, neither a standard terminology, except the one of Auer and Pillar (2020). They described only epithallial cells and for the *Lithothamnion* type they use the term block. This term is in our opinion far from being explicative and/or appropriate from the point of view of morphology. The term "block" does not express any type of characterization in term of shape.

P5L160, it needs to be shown how the growth rate can be calculated; according to the description in P1L55, is not the reason for the band formation still under argument? If so, wouldn't the growth rate be based on the "assumption" that the band is one year old?

We will add that our banding represents band of type 3 and 4 (Foster, 2001), representing one year, also in the Result section, and not only on the Discussion. Moreover, we will add the details on how we calculated the growth-rate.

P5L169 If secondary crystals were there, how the samples were stored. Describe the conditions from the time of collection to the time of observation.

We will add all the requested details to clarify our finding and support our interpretation. At this stage, we will be more conservative in defining our observation.

P7L224 The reviewer thinks it would be easier to convey the idea if authors make a comprehensive schematic diagram of the microstructure patterns that are not affected by the environment.

The microstructure pattern that is not affected by the environment are exposed in figure 6. We would be grateful to have further indications about the type of possible additional diagrams

Fig.8 c): The black arrows are hard to see. Please border the arrows with white.

Ok. To indicate with the same color PW and SW across figures we will change the color and make them as more possible visible.

Fig.8 h) The black arrows are hard to find. Please border the arrows with white.

Ok. To indicate with the same color PW and SW across figures we will change the color and make them as more possible visible.